# Consumer Willingness to Pay for Food Products Enriched with Brewers’ Spent Grain: A Discrete Choice Experiment

**DOI:** 10.3390/foods13223590

**Published:** 2024-11-10

**Authors:** Cinu Varghese, Patricia Arcia, Ana Curutchet

**Affiliations:** 1Latitud—Laboratorio Tecnológico del Uruguay (LATU Foundation), Montevideo 11500, Uruguay; getcinuvarghese@gmail.com (C.V.); parcia@latitud.org.uy (P.A.); 2Engineering Department, Universidad Católica Del Uruguay, Montevideo 11600, Uruguay

**Keywords:** product category, sustainability logo, by-product, consumer perception

## Abstract

Brewers’ spent grain (BSG), a nutrient-rich by-product, offers the food industry a sustainable opportunity. This study explores consumer willingness to pay (WTP) for food products enriched with BSG, focusing on the influence of sustainability logos and brand information. Using a discrete choice experiment (DCE), analyze how these attributes impact consumer preferences for two products: BSG-enriched bread and chocolate dessert. Key variables included the presence of sustainability logos and BSG information, brand type (premium, low-cost, or no-brand), and price. An online survey was conducted, and the multinomial logit (MNL) model was applied to the data (n = 402). Overall, these results suggest that sustainability logos and BSG information positively influence consumer choices, although brand significance varies across product categories. For bread, the brand plays a critical role in purchasing decisions, while for chocolate dessert, the price is the main decision factor. This research highlights that through the addition of BSG, the bread and chocolate manufacturing industry in Uruguay can increase profits with a premium price and improve product quality, transforming the food industry and advancing sustainable development.

## 1. Introduction 

Brewers’ spent grain (BSG) is the primary by-product of the brewing industry, consisting mainly of the residual malt and grain husks left after the extraction of wort during the brewing process. [1]. For every kg of beer produced, 0.2 kg of wet BSG has been reported to be generated [2]. According to the FAO, in 2020, global beer production was estimated at 210 million tons [3], which is equivalent to 42 million tons of BSG. The use of BSG represents opportunities for the food industry to reuse by-products due to its fiber, protein, and mineral-rich composition, low cost, and wide range of functional ingredients [4]. The use of by-products is increasingly recognized as a critical strategy to achieve sustainable production and reduce environmental impact. By changing what would be wasted into valuable resources, industries can improve resource efficiency, lower disposal costs, and contribute to the circular economy [5]. The use of by-products not only minimizes the environmental footprint of production processes but also creates new revenue streams and is in line with the growing consumer demand for eco-friendly products. Nyhan et al. [6] identified 19 food manufacturers operating in the US, Europe (Ireland, France, Germany, Denmark, Switzerland, UK) and Asia-Pacific (New Zealand, Australia, Canada, India) that incorporate dried BSG into food products. The potential of BSG is shown in several companies that are actively involved in processing brewers’ spent grain (BSG) into high-value food products. As the incorporation of BSG into food products consolidates, the evaluation of willingness to pay provides essential information for market positioning, pricing strategies, and product development, particularly in response to consumer preferences for sustainable and trusted offerings. 

Willingness to pay (WTP) is defined as the maximum amount of money an individual is willing to spend to acquire a good or service or to avoid something undesirable. Understanding what consumers value at the time of purchase and which attributes ultimately determine their choice is to determine their willingness to pay. It reflects the perceived value of a product or service to the consumer and is a crucial metric in assessing consumer preferences and demand. WTP is often used in economics and marketing to inform pricing strategies, cost–benefit analysis, and policy decisions [7]. Consumer WTP can be affected by many factors, such as country, product category, and socio-demographic factors (sex, age, income, and education) [8]. Understanding consumers’ willingness to pay for products perceived as sustainable is crucial for both market positioning and product development strategies. The WTP of sustainable products is often influenced by factors such as environmental awareness, personal values, and trust in sustainability claims, making it a dynamic and context-specific measure [9]. 

The world’s paradigm of willingness to pay for sustainable products and services is changing significantly; 85% of the world’s population has bought sustainable products over the last 5 years, and half of them rank sustainability among their top five value drivers at the time of purchase, and more than a third (34%) are willing to pay more for sustainable products and services [10]. In a recent meta-analysis conducted by Li and Kallas [11], gender, region, sustainable attributes, and food categories all influenced consumers’ willingness to pay for sustainable food products. Moreover, there is a significant regional variation in consumers’ willingness to pay.

Most existing studies investigating the WTP for various food attributes in the food sector focus on highly flexible food categories such as meat, fruit, and vegetables [12,13,14,15,16,17]. There is, however, a noticeable gap in quantitative analyses of consumer WTPs for food attributes, particularly sustainability logos and BSG information. De Magistris et al. [18] explored this behavior through a real choice experiment that focused on consumer preferences for organically and locally grown almonds, finding that consumers are more likely to pay a higher price for products with sustainability attributes, especially those produced locally or labeled organic. Stelick et al. [19] study the impact of sustainability and nutritional messaging on Italian consumers’ purchasing intention for cereal bars made with BSG. A significant positive effect on purchase intent was observed when providing nutritional (fiber content) or sustainability (use of upcycled ingredients) information, although the optimal price point for the BSG was lower than the control. G.C. Miranda-de la lama et al. [20] reported segmentation of Mexican consumers for welfare-friendly products based on their perception of animal welfare attributes and willingness to pay. Curutchet et al. [21] studied the effect of the way of communication of BSG enrichment on the consumer response to beef burgers. They found that consumers wanted to incorporate BSG into the products and wanted to know the origin of the ingredients but did not want to know more about the by-product.

Food attributes can be classified as intrinsic or extrinsic [22]. Extrinsic attributes include food labels, product types, and brand while intrinsic attributes refer to taste, quality, and nutritional value. The labeling of by-products in food products is a critical factor that influences consumer acceptance and marketability. Consumers prefer detailed information about the by-product and should improve purchase intention. Effective communication strategies through labels can increase consumer acceptance of more convenient, healthier, and environmentally friendly products [23]. When by-products are transparently communicated as ingredients, consumers often respond with skepticism, associating them with lower quality or inferior taste [24]. However, strategic labeling that highlights the sustainability, nutritional benefits, or innovative aspects of the use of by-products can positively change consumer perceptions. E. J Van Loo et al. [8] reported that Belgian consumers valued free-range claims in terms of willingness to pay, attracting premiums ranging from 43% to 93%. These are followed by an animal welfare label, a carbon footprint label, and lastly, organic logos. P. Tait et al. [25] stated that the presence of sustainable attributes, especially price, can significantly influence the consumers’ willingness to pay. Another extrinsic attribute is that the brand plays an important role in modeling different consumer perceptions. Brands are built to establish associative bonds in the consumer’s memory and to provide product experiences, above all, generating a climate of trust among consumers [26]. Tian et al. [27] studied the preference of the consumer brand over infant formula and stated that product characteristics and external environmental factors could influence consumer brand preferences.

In this context, to finally choose among multiple alternatives, discrete choice experiments (DCE) are often used research methods in food research because of their ability to uncover trade-offs. The DCE method aims to uncover the attributes that drive the decision-making of food producers and consumers on different topics [28]. In a discrete choice experiment, participants are asked to choose between hypothetical alternatives described by a set of attributes with varying levels. DCEs offer several advantages compared to observational data from population surveys or sales transactions. They directly capture the participants’ choices between different alternatives [29]. 

Although estimates of consumer WTPs for sustainable products have received considerable attention in developed markets, little is known about Lain American countries, especially emerging economies such as Uruguay. Empirical estimation of consumer willingness to pay (WTP) for sustainability attributes of BSG products is scarce. In light of the aforementioned facts, this research aims to contribute to the understanding of how the inclusion of by-products like brewers’ spent grain and the strategic use of labeling such as sustainability logos and BSG information and branding influence consumer purchasing decisions and willingness to pay for sustainable food products using the Discrete Choice Experiment. From this, we propose the following working hypotheses: 

**Hypothesis** **1 (H1).**
*Food products with sustainability logos and BSG information have an effect on purchase intention (H1a) and willingness to pay (H1b).*


**Hypothesis** **2 (H2).**
*Brand has an effect on purchase intention (H2a) and willingness to pay (H2b) for sustainable products.*


**Hypothesis** **3 (H3).**
*The category product has an effect on the intention to purchase (H3a) and willingness to pay (H3b).*


## 2. Materials and Methods

### 2.1. Discrete Choice Experiment Design

#### 2.1.1. Selection of Attributes and Levels

In a discrete choice experiment, consumers will be asked to complete a series of choice sets of products consisting of several alternatives, each described by their characteristics. The consumer will choose a preferred product profile from each set of choices [29]. In this study, two products were considered: BSG-enriched bread and salchichón de chocolate, a famous Uruguayan chocolate dessert. As bread and salchichón de chocolate play such an important cultural and dietary role in Uruguay, analyzing consumer preferences and purchasing behavior around both products can provide valuable insight into broader consumption patterns and socioeconomic trends. 

The choice of 3–5 attributes and 10–15 variables (levels) to be optimal for a bias-reduced study [30]. Three attributes, such as (i) sustainability logo with brewery spent grain (BSG) information, (ii) brand information, and (iii) price with different levels, were included (Table 1 and Table 2). Climate change has led to calls for labeling to allow consumers to differentiate between more or less sustainable options. Such calls assume that when consumers receive the appropriate information on the label, they are expected to change their purchases and make more sustainable choices [31]. Therefore, for this study, we first included the sustainability logo and the BSG information because our previous study on consumer attitudes towards BSG-enriched products under informed conditions showed the importance of the sustainability logo and the BSG information in consumers’ purchase intention [21]. Therefore, we consider BSG information and the sustainability logo together as a single attribute in this study. The BSG information provided on the label was “Did you know that the brewer’s spent grain is a coproduct generated during the beer-making process? More than half a kilogram of it is produced for every liter of beer. Although it is generally not used for food production, its high fibre and protein content make it a highly nutritious ingredient”.

Second, we include brand, as branding is an important factor and extrinsic attribute that signals quality and individual trust in consumer purchase decisions [32]. Three brands for each product were selected: a premium brand, a low-cost brand, and a no-brand option. The selection of brand would depend on the consumer’s purchase in the grocery store. In the case of the no-brand option, the labels were designed for exclusive use in this investigation by Syrah Design Studio. Lastly, we include price because this study aims to estimate the willingness to pay for BSG-enriched products, so the monetary cost must also be included as an attribute. Therefore, four price levels were selected based on the current market price of bread and chocolate dessert in Montevideo, Uruguay. This study used an online survey, which limited its ability to evaluate sensory characteristics of BSG-enriched products. As sensory attributes influence consumer perceptions and acceptance, excluding them may have limited the depth of understanding of consumer preferences.

#### 2.1.2. Experimental Design

The choice sets were designed using the choice design platform in SAS JMP Pro 17 software. To maximize the precision of the parameter estimate, the choice sets were designed using a D-efficient Bayesian design using the selected attributes and their levels. The choice set was designed for the pilot study and conducted with 30 consumers to collect preliminary data, and they were not considered in the final analysis. The experimental design for the pilot study consists of two blocks (surveys) with 15 respondents each. Each choice task consisted of two product alternatives (options A and B) and an opt-out option (option C). Figure 1 and Figure 2 show an example of a choice set used in the study for bread and chocolate dessert, respectively. In this study, eight sets were used for each consumer to prevent fatigue of the respondents. The pilot study evaluated whether respondents had understood the experiment and could handle the eight-choice tasks. In addition, it was asked about the suitability of the attributes and levels used in the experiment and whether any modifications were needed before the final design. The results from the pilot study were used to define the final survey and inform the priors of the Bayesian D-efficient design. The main advantage of Bayesian D-efficient design is that it avoids choosing a set of profiles that completely dominate the other profiles of all attributes, using the appropriate prior distribution [33]. 

#### 2.1.3. Survey

The pilot study and the final survey were conducted in Spanish using Qualtrics^TM^ software (2023, Provo, UT, USA). The choice experiment was introduced to consumers with an explanation of the research and a clear description of the attributes and levels. Information on the logo and the different brands used was displayed before the choice task to help select the choice set. To reduce potential hypothetical bias that can affect consumer WTP estimates in stated preference studies, respondents read a cheap talk (CT) script before performing choice tasks [22]. Through this method, the respondent can self-correct the hypothetical bias issue. In addition to the choice task, respondents completed questionnaires about their sociodemographic status, habits, and attitudes. To ensure the quality of the data, we included only participants who had said “They had given their full attention to the questions so far, and in their honest opinion, they believed that we should use their responses for the study”. This question has strategically been placed before the choice tasks. 

### 2.2. Multinomial Logit (MNL) Model

The multinomial logit (MNL) model relies on random utility theory [34]. The basic assumption of the theory of random utility is based on the hypothesis that individuals act rationally by choosing the alternative with the greatest utility. Thus, if the utility offered by such an alternative is the highest among the various options, the probability of selecting a given alternative will be higher [8]. “A decision maker i faces J alternatives and the utility derived from choice alternatives j is U_ij_, j = 1, …, J. The ith consumer’s utility of choosing option j includes a systematic component Vij and a random or unobservable component ε_ij_”.
(1)$$U_{ij}=V_{ij}+\varepsilon_{ij}$$

When we consider the random errors in Equation (1) as independently and identically distributed among the J alternatives and N consumers, the probability that consumer i selects the alternative j from a set of choices, Ci, can be given by the MNL model as follows:(2)$${Prob}_{i}\left\{j\;chosen\right\}=\frac{e^{V_{ik}}}{\sum_{k=1}^{j}e^{V_{ik}}}with\;k\;\epsilon \;Ci$$

When the total utility of the buyer is the linear sum of the utility given by the different attributes and prices of the products. Thus, the formula for the consumer’s WTP is as follows: WTP_i_ = −β_i_/βp(3)

WTP_i_ is the willingness to pay for the i − h attribute level, β_i_ is the mean estimation coefficient of that attribute level, and βp is the estimation coefficient of price.

### 2.3. Data Analysis

Data collected through DCE enable quantitative and statistical interpretation of the relative importance of attributes and attribute levels in assessing the priority classification of the selection of BSG-enriched products. The data are analyzed by estimating the utility function in the MNL model using a maximum likelihood approach. The overall significance of the attributes was estimated with likelihood ratio (LR) tests, and the relative importance of the attributes was measured by the log *p* value. The entire data were analyzed using the Choice Modeling platform in the statistical software package SAS JMP 17.

## 3. Results and Discussion

### 3.1. Sociodemographic Analysis

A total of 402 respondents completed the survey, and the demographic analysis is shown in Table 3. The gender distribution is similar to the census population (INE, 2023). Regarding age, 30% of the respondents were 18 to 35 years old, 47% were 35 to 64 years old, and 20% were 65+ years old. The sample was well balanced regarding sex and age, while rather highly educated compared to the general population. 37% of the respondents completed university, and 40% worked in the private sector. The household income of 37% of respondents ranged from UYU 60,000 to UYU 130,000. The sample is comparable in terms of sex, age, education, employment, and income. The selected sample is generally considered to represent our target population. 

### 3.2. Estimated Parameters for Multinomial Logit Model (MNL) 

#### 3.2.1. BSG Enriched Bread

Figure 3 and Table 4 show that all three attributes, including the opt-out option, are statistically significant (*p* < 0.05) contributors to consumer preference, meaning that none of them is considered irrelevant to the priority setting. According to the results, the most influential attributes for bread selection (based on the LR test) are brand (LR χ^2^ = 115, *p* = 0.0005) and the price (relative importance 92%), which is almost equal in importance. BSG information with a sustainability logo has the least influence. Bread is an important food in the Uruguayan diet and is eaten every day, served to the family, and purchased by households. Due to this cultural importance, consumers can attribute safety and familiarity to familiar brands as indicators of quality and credibility. After brands, price plays an important role, reflecting the frequency of bread purchases and the need to balance quality with affordability. Even if a sustainability logo is considered valuable, it is ranked third, indicating that sustainability can only be a secondary factor in daily essentials such as bread. The importance of the brand is consistent with the findings of numerous previous studies showing the influence of the brand on consumer preferences [35,36]. 

Zahid H. and Hafeez [37], in their work on the effect of brand image on consumer taste preferences, reported that there is a significant relationship between brand image and the brand currently preferred. These findings were corroborated by Abdurrahman and Yasar [38], who found that the brand name of a product has a significant impact on the general preferences of Turkmenistan consumers.

The results of the estimation of the parameters of the MNL model for bread are shown in Table 4. As can be seen in Table 4, the estimate of the opt-out option was negative and significant (estimate: 1.2852, *p*-value: 0.00), indicating that consumers tend to select one of the two product alternatives in a set as opposed to the ‘opt-out’ option. All estimates that hold a more positive marginal utility translate to a preferable product profile. Concerning the labeling of the BSG information and the sustainability logo, the consumer shows a positive preference with a positive marginal utility value of 0.28 for the presence of BSG information and the sustainability logo. The strong preference for the BSG information and sustainability logo indicates the awareness of consumers on health and environmental aspects [39]. Regarding brand, the premium brand with a marginal utility of 0.56 shows the consumer prioritizes the premium brand over low cost and no brand images. These results support Irma Tikkanen Mari Vääriskoski’s theory [40] that the market position of premium brands is to provide consumers with high value-added products with innovative designs and, sometimes, higher quality products than national brands. By linking the social image of the product, the consumer shows more trust in premium bread. On average, for bread, the results show that consumers prefer to choose low-priced UYU 125 (estimate: −0.01), premium brand BSG-enriched bread labeled with BSG information with a sustainability logo. The collected data are consistent with those of De Boni et al. [41], who have highlighted consumer sensitivity to prices, deep knowledge of sustainability, and a positive attitude toward the consumption of special bread, which plays a role in explaining the variability in bread choices.

#### 3.2.2. BSG Chocolate Dessert

Interestingly, compared to bread, we can notice a different ranking of the attributes of consumer preference for the BSG chocolate dessert. As can be seen in Figure 3 and Table 5, for the BSG chocolate dessert, the price was the most important attribute, followed by the BSG information and the sustainability logo (relative importance 75). The brand was much less influential, with a relative importance of 30% compared to the price (Figure 3). Salchichón de chocolate (chocolate dessert) is a popular and traditional dessert that is occasionally consumed in Uruguay. As a result, consumers tend to consider the price as the first consideration when it comes to discretionary items. An increasing sense of environmental sensitivity appears to be evident in the preference for sustainability logos over brands. It seems that Uruguayans have become more conscious of sustainable choices, even when it comes to traditional treats. These results are in line with those obtained by Kiss et al. [42], who studied the decision to buy chocolate bars from the Hungarian population. The authors indicated that the increase in chocolate prices led to a decrease in consumer utility, reflecting the sensitivity to buyer prices. This result has an important impact on the pricing strategy of the Hungarian chocolate industry.

In Table 5, the estimates for the chocolate dessert are shown, representing the marginal utilities assigned to the different attribute levels. The estimates showed that consumers preferred to choose low-priced (UYU 225), low-cost brands (marginal utility: 0.156), chocolate desserts labeled with BSG information, and sustainability logos (marginal utility: 0.277). The pattern found here corresponds to that of numerous previous findings supporting the crucial role of price in chocolate decision-making [42,43,44,45,46,47]. The positive marginal utility of selecting the BSG information and sustainability logo is consistent with previous studies indicating that the presence of sustainability labels could create a positive effect and influence consumer purchasing options [48]. Regarding brands, both low-cost and premium brands show a positive marginal utility, and consumers prefer the low-cost brand over premium brands, which is different from the findings observed by Lybeck et al. [43], where consumers are more loyal to the manufacturer’s brand than store-brand products. For the consumer who preferred the opt-out option, the attribute combination that led to choosing the opt-out option was a high-priced and premium brand with no BSG information and a sustainability logo. 

### 3.3. Willingness to Pay for Different Attributes

Table 6 and Table 7 show the willingness of the consumer to pay for different attributes in BSG-enriched bread and chocolate dessert. Based on the estimated parameters of the MNL model presented above, the consumer’s WTP for the attributes of BSG information with sustainability logo and brand. The baseline pay was fixed to UYU 125 for bread and UYU 220 for chocolate dessert, along with no brand and no BSG information and sustainability logo, and the percentage premium was calculated with the baseline pay. Consumers were willing to pay a higher price for bread with the BSG information and sustainability logo (UYU 174, or 39% higher) and a product branded premium (UYU 204.49, or 63.6% higher). Additionally, the price change for low-cost bread was only UYU 7.06 (5.6%). For BSG chocolate dessert, consumers were willing to pay UYU 287 for chocolate dessert labeled with BSG information with the sustainability logo. The willingness to pay for low-cost and premium brands was almost similar, with a premium rate of 123.1% and 122.3%. Independent of the product, consumers are willing to pay a premium for both bread and chocolate dessert. 

These findings confirm those of earlier studies, reported by S. Li and Z. Kallas [11] through meta-analysis, indicating that a growing trend has been observed in the average premium consumers worldwide willing to pay for sustainable food products. Alsubhi et al. [49] similarly conclude from their data that consumers prefer healthier food products and are willing to pay more for healthier alternatives, demonstrating consumer preference and therefore the potential demand for healthier options. 

The results are in agreement with studies conducted by Valenzuela, L. et al. [50], who studied Chilean consumer preference and willingness to pay for sustainable wine. The authors highlighted that 22% of consumers were willing to pay a premium price for organic wines, while 19% expressed a willingness to pay premium prices for sustainable wines. G.C. Miranda-de la Lama et al. [20] reported that Mexican consumers were willing to pay more for welfare-friendly products that were properly certified. Medeiros et al. [51] studied Brazilian consumers perceived value of green products and identified that although the perceived value of green products increases willingness to pay in the purchasing decision, explicit attributes related to environmental sustainability were deemed less relevant during a typical purchase decision process. Gao et al. [52] reported that the willingness of the Chinese consumer to pay for sustainable milk is 40% higher than conventional milk. Fang et al. [53] reported a similar trend in Chinese consumers who were willing to pay premium rates that exceeded 100% for superior taste quality, organic certification labels, and green certification labels for rice. Another study reported that most Spanish consumers were willing to pay a higher price for sustainable wines [54]. Consistent with the present findings, Arama et al. [55] reported that the integration of cricket flour into existing market-driven consumer-familiar food products significantly increased acceptability and willingness to pay. A substantial premium rate reflects the strong desire to invest in BSG-enriched products with delicious and healthy attributes, reflecting both the inherent value and the equilibrium of the market between supply and demand [56,57].

It is well established that WTP for sustainable products has been extensively studied in developed markets; however, this is not the case in Latin America, particularly for staple food categories instead of more flexible ones like meat, fruits, and vegetables [12,13,14,15,16,17]. Few studies are reported that study the consumer preferences for sustainable products from Latin America [20,50,51]. This study provides both theoretical and practical insights into consumer perceptions of sustainability logos and BSG information in Uruguay. Previous research has consistently highlighted the influence of sustainability attributes on consumer purchasing decisions [11,20]. Our results corroborate these findings, demonstrating that sustainability logos and brewers’ spent grain (BSG) information significantly impact consumer willingness to pay (WTP). Theoretically, this reaffirms the theory that sustainability-oriented marketing features can add value to food products, as suggested by Van Loo et al. [58] and Zahid and Hafeez [35], who found that such attributes enhance consumer perception of quality and trust, ultimately affecting their purchasing behavior. In practical terms, our study provides useful tips for marketers and product developers. It highlights how important it is to clearly explain the environmental and health benefits of BSG-enriched products. This can help increase consumer willingness to pay and promote a more sustainable food market. This approach not only meets the growing consumer demand for eco-friendly products but also aligns with global sustainability goals by reducing food waste and enhancing the circular economy. This study helps to strengthen the framework for understanding sustainability preferences globally and highlights emerging market opportunities by comparing our findings to those from developed economies.

## 4. Conclusions 

In the study, a discrete choice experiment was conducted to evaluate consumer preferences for bread and chocolate dessert enriched with brewery spent grain (BSG) in various attributes: BSG information and sustainability logo, brand, and price. Taken together, these findings are in line with our expectations as well as with theoretical and empirical experiences. The estimations of the MNL model in our research support the idea that the presence of BSG information with sustainability logo (H1) and brand (H2) affects purchase intention and willingness to pay. Preference and willingness to pay for bread and chocolate dessert were different, and the important attributes for the selection of both products were different. These results support our third hypothesis (H3), which categories of products have an effect on purchase intention and willingness to pay. The presence of sustainability information, particularly through BSG enrichment and logos, significantly enhances consumer willingness to pay across different food categories. Finally, the findings of this work suggest that BSG-enriched food products can be a win-win for both consumers and the environment, offering a sustainable and desirable option. With a higher WTP, the bread and salchichon manufacturing industry in Uruguay can improve profits through the addition of BSG and improve product quality, transforming the food industry and advancing sustainable development. 

As a limitation of this study, the study relied on hypothetical choices or normal purchasing behavior, which can introduce hypothetical bias despite the inclusion of a cheap talk script. Furthermore, the discrete choice experiment was conducted with a small, geographically localized sample (Montevideo), which may limit the generalizability of the findings to other populations. Future research could expand the sample size, include more diverse demographic groups, and incorporate real-world purchasing data to validate and extend these findings. The MNL model does not consider the heterogeneity of the population. The Mixed Logit (ML) model overcomes this limitation. Therefore, this study does not assess how sociodemographic characteristics, such as education, income, and age, can influence willingness to pay for these attributes. Future research could expand on our findings with ML models and alternative segmentation methods (e.g., latent class modeling) examining how sociodemographic factors influence willingness to pay for sustainable products. This would provide more information on the consumer segments most responsive to sustainability and help tailor marketing strategies for diverse groups. Therefore, future studies could be conducted with sensory variables such as texture, taste, and flavor, which may also be important determinants of consumer WTP.

## Figures and Tables

**Figure 1 foods-13-03590-f001:**
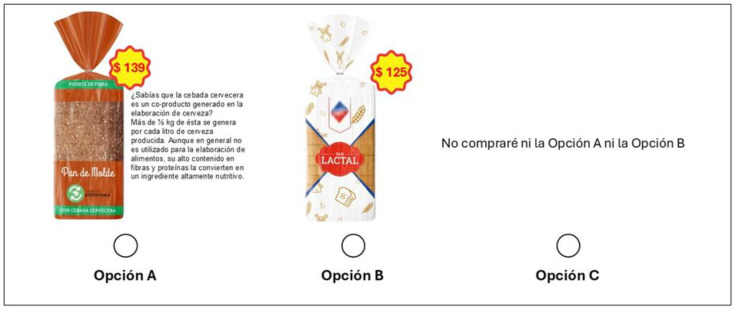
Example of the choice set used for the BSG bread in the survey.

**Figure 2 foods-13-03590-f002:**
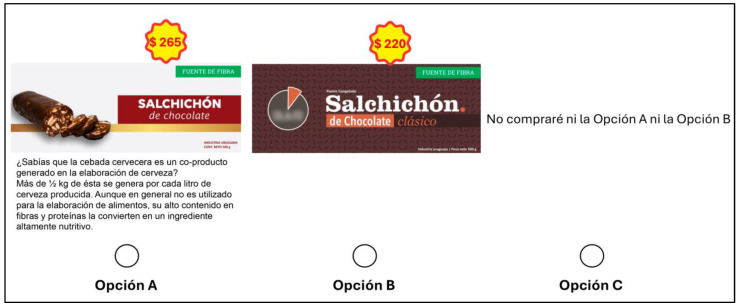
Example of the choice set used for the BSG chocolate dessert in the survey.

**Figure 3 foods-13-03590-f003:**
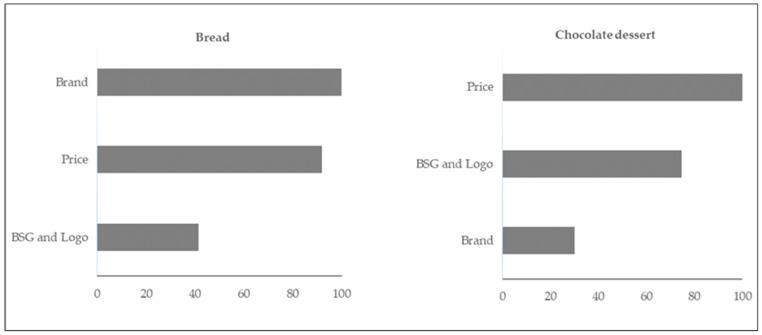
Importance of the three attributes to the purchase intention relative to the most important attribute ‘Brand’ (bread) and ‘Price’ (chocolate dessert), which is set at 100.

**Table 1 foods-13-03590-t001:** Attributes and levels of BSG bread.

Attributes	Levels
BSG information and sustainability logo	No
	Yes
Brand	No brand
	Low cost
	Premium
Price (UYU)	125
	139
	175
	229

**Table 2 foods-13-03590-t002:** Attributes and levels for BSG chocolate dessert.

Attributes	Levels
BSG information and sustainability logo	No
	Yes
Brand	No brand
	Low cost
	Premium
Price (UYU)	220
	265
	300
	330

**Table 3 foods-13-03590-t003:** Consumer sociodemographic characteristics.

Sociodemographics	N = 402	
Gender		
Female	210	52%
Male	192	48%
Prefer not to say	0	0
Age		
18–25	70	17%
26–35	58	14%
36–45	54	13%
46–55	79	20%
56–65	58	14%
66–75	49	12%
76–85	34	8%
Education		
Primary school	3	1%
Secondary/middle school	106	26%
High school/college	141	35%
University degree	148	37%
Others	4	1%
Employment		
Student	54	13%
Independent worker	50	12%
Private sector worker	194	48%
Public sector worker	39	10%
Retired	40	10%
Unemployed (seeking work)	8	2%
Not in paid employment	4	1%
Others	13	3%
Income		
<UYU 60,000	49	12%
UYU 61,000 to 130,000	130	32%
UYU 131,000 to 260,000	116	29%
>UYU 261,000	48	12%
Not declared	59	15%

**Table 4 foods-13-03590-t004:** Multinomial logit estimates and significance of the attribute effects obtained from likelihood ratio (LR) tests for BSG Bread.

	Estimate	Lower 95%	Upper 95%	L-R ChiSquare	Marginal Utility	*p* Value
Bsg and logo				43.54		<0.001
No	−0.271	−0.3531	−0.1903		−0.28	
Yes	0.271 *	0.1904	0.3531		0.28	
Brand				115.58		<0.001
No	−0.3172	−0.4199	−0.2153		−0.32	
Low cost	−0.2395	−0.3458	−0.1342		−0.24	
Premium	0.5567 *	0.4199	0.2153		0.56	
Price	−0.0109	−0.0131	−0.0088	100.97		<0.001
Opt-out	−1.2852	−1.6428	−0.9301	51.13		<0.001
* AIC	4304.9					
* BIC	4335.5					
−2 LogLikelihood	4294.9					
−2 Firth LogLikelihood	4255.9					

* AIC: Akaike’s information criterion. * BIC: Bayesian Information Criterion * Coefficient estimates corresponding to the last level of an attribute are calculated as minus the sum of the estimates for the other levels of that attribute.

**Table 5 foods-13-03590-t005:** Multinomial logit estimates and significance of the attribute effects obtained from likelihood ratio (LR) tests for chocolate dessert.

	Estimate	Lower 95%	Upper 95%	L-R ChiSquare	Marginal Utility	*p* Value
Bsg and logo				75.27		<0.001
No	−0.297	0.0345	−0.3649		−0.297	
Yes	0.297 *	0.3649	−0.0345		0.297	
				32.36		<0.001
No	−0.2929	0.0526	−0.3965		−0.292	
Low cost	0.1557	0.0473	0.0628		0.156	
Premium	0.1372 *	−0.0526	0.3965		0.137	
Price	−0.0088	0.0008	−0.0104	102.07		<0.001
Opt-out	−1.8367	0.24113	−2.3101	58.83		<0.001
* AIC	4884.11					
* BIC	4915.37					
−2 LogLikelihood	4874.10					
−2 FirthLogLikelihood	4834.18					

* AIC: Akaike’s information criterion. * BIC: Bayesian Information Criterion * Coefficient estimates corresponding to the last level of an attribute are calculated as minus the sum of the estimates for the other levels of that attribute.

**Table 6 foods-13-03590-t006:** Willingness to pay for BSG bread with different attributes.

Attributes	Level	WTP (UYU)	Std. Error	Lower 95% (UYU)	Upper 95% (UYU)	Premium Rate (%)
BSG and logo	Yes	174.39	0.32214	48.76	50.02	139.5
Brand	Low	132.06	0.38798	6.30	7.82	105.6
Brand	Premium	204.49	0.48268	78.55	80.44	163.6

**Table 7 foods-13-03590-t007:** Willingness to pay for BSG chocolate dessert with different attributes.

Attributes	Level	WTP (UYU)	Std. Error	Lower 95% (UYU)	Upper 95% (UYU)	Premium Rate (%)
BSG and logo	Yes	287.88	0.56514	66.78	68.99	130.5
Brand	Low	271.25	0.6146	50.05	52.46	123.1
Brand	Premium	269.13	0.58059	48.00	50.27	122.3

## Data Availability

The original contributions presented in the study are included in the article, further inquiries can be directed to the corresponding author.

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
