# Peer review of "Consumer Willingness to Pay for Food Products Enriched with Brewers’ Spent Grain: A Discrete Choice Experiment"

_foods, 2024, doi:10.3390/foods13223590_

Round 1
Reviewer 1 Report
Comments and Suggestions for Authors
See comments

Author Response
Reply reviewer 1 in file attached.

Reviewer 2 Report
Comments and Suggestions for Authors
This paper is wel written. but there are several limitations.
1. The introduction is poor. The theoretical contribution is not described at all.
2. The development of research hypotheses should be established more robustly in a new section.
3. I expected to test your hypotheses using the logit model, but your analysis is somewhat different.
4. The section of conclusion is very poor.
Author Response
Reply reviewer 2 attached in a separate file.

Round 2
Reviewer 1 Report
Comments and Suggestions for Authors
Thank you for the responses to the review. The work responds to the observations raised.
Author Response
Thank you very much for you revision. We are attaching a new version with reviwers 2 comments.
Regards,
Ana Curutchet

Reviewer 2 Report
Comments and Suggestions for Authors
The overall correction is fine. However, the discussion with theoreitcal and practical implications should be robustly addressed based on the comparison of prior studies.
Author Response
Thank you for the overall evaluation of our work and for allowing us to review it. We have modified the manuscript accordingly, and the detailed corrections are listed below.
Comment 1
The overall correction is fine. However, the discussion with theoreitcal and practical implications should be robustly addressed based on the comparison of prior studies.
Response 1
In response to your feedback, we have revised the discussion section of our manuscript to incorporate a comprehensive comparison with prior studies. We have included references that work in the influence of sustainability attributes on consumer purchasing decisions and discuss how our findings both corroborate and expand upon existing research. The following new paragraph is added under results and discussion section and highlighted (yellow) in the revised manuscript for better clarity (Page no. 12, line 410-430).
Results and discussion (Excerpt)
It is well established that WTP for sustainable products has been extensively studied in developed markets; however, this is not the case in Latin America, particularly for staple food categories instead of more flexible ones like meat, fruits, and vegetables [48-53]. Few studies are reported which study the consumer preferences for sustainable products from Latin America [56,60-61]. This study provides both theoretical and practical insights into consumer perceptions of sustainability logos and BSG information in Uruguay. Previous research has consistently highlighted the influence of sustainability attributes on consumer purchasing decisions [40,56]. Our results corroborate these findings, demonstrating that sustainability logos and Brewers' Spent Grain (BSG) information significantly impact consumer willingness to pay (WTP). Theoretically, this reaffirms the theory that sustainability-oriented marketing features can add value to food products, as suggested by Van Loo et al. [57] and Zahid and Hafeez [21], who found that such attributes enhance consumer perception of quality and trust, ultimately affecting their purchasing behavior. In practical terms, our study provides useful tips for marketers and product developers. It highlights how important it is to clearly explain the environmental and health benefits of BSG-enriched products. This can help increase consumer willingness to pay and promote a more sustainable food market. This approach not only meets the growing consumer demand for eco-friendly products but also aligns with global sustainability goals by reducing food waste and enhancing the circular economy. This study helps to strengthen the framework for understanding sustainability preferences globally and highlights emerging market opportunities by comparing our findings to those from developed economies.
Regards,
Ana Curutchet
